# Microtubule-dependent ribosome localization in *C. elegans* neurons

Kentaro Noma[1,2†‡], Alexandr Goncharov[1,2], Mark H Ellisman[3], Yishi Jin[1,2]*

[1]Division of Biological Sciences, Neurobiology Section, University of California, San Diego, San Diego, United States; [2]Howard Hughes Medical Institute, University of California, San Diego, San Diego, United States; [3]National Center for Research in Biological Systems, Department of Neurosciences, School of Medicine, University of California, San Diego, San Diego, United States

**Abstract** Subcellular localization of ribosomes defines the location and capacity for protein synthesis. Methods for in vivo visualizing ribosomes in multicellular organisms are desirable in mechanistic investigations of the cell biology of ribosome dynamics. Here, we developed an approach using split GFP for tissue-specific visualization of ribosomes in *Caenorhabditis elegans*. Labeled ribosomes are detected as fluorescent puncta in the axons and synaptic terminals of specific neuron types, correlating with ribosome distribution at the ultrastructural level. We found that axonal ribosomes change localization during neuronal development and after axonal injury. By examining mutants affecting axonal trafficking and performing a forward genetic screen, we showed that the microtubule cytoskeleton and the JIP3 protein UNC-16 exert distinct effects on localization of axonal and somatic ribosomes. Our data demonstrate the utility of tissue-specific visualization of ribosomes *in vivo*, and provide insight into the mechanisms of active regulation of ribosome localization in neurons.

DOI: https://doi.org/10.7554/eLife.26376.001

*For correspondence:
yijin@ucsd.edu

Present address: †Group of Nutritional Neuroscience, Neuroscience Institute, Graduate School of Science, Nagoya University, Nagoya, Japan; ‡JapanScience and Technology Agency, Saitama, Japan

Competing interests: The authors declare that no competing interests exist.

## Introduction

Localization of organelles defines the local environment and functionality of cells. Ribosome localization determines the local capacity for protein synthesis, which is particularly important in polarized cells, such as neurons. The observation of polysomes in the base of dendritic spines in rat dentate gyrus neurons provided crucial early evidence for local translation in neurons (*Steward and Levy, 1982*). Aberrant local protein synthesis disrupts neuronal plasticity in models for neurodevelopmental disorders, such as Tuberous sclerosis (TSC) and Fragile X syndrome (FXS) (*Auerbach et al., 2011*). Furthermore, the importance of local translation is highlighted by unique localization of specific mRNAs. For example, $Ca^{2+}$/calmodulin-dependent protein kinase II α(CaMKIIα) mRNA is localized to dendrites via a *cis* regulatory element in its 3'-UTR, and removal of this element impairs long-term potentiation (LTP) and spatial memory (*Miller et al., 2002*); β-actin mRNA is localized to the growing tip of an axon, the growth cone, where its polarized translation is important for growth cone turning towards an attractant (*Leung et al., 2006*; *Yao et al., 2006*). Despite their potential impact on local translation, mechanisms underlying ribosome localization *in vivo* are largely unexplored, partly due to the limited tools to visualize ribosomes in live multicellular organisms. One striking example of ribosome localization is that dendritic ribosomes associated with the netrin receptor, Deleted in Colorectal Cancer (DCC), are stalled and inactive, and released upon netrin signaling (*Tcherkezian et al., 2010*).

Electron microscopy (EM) studies have long provided essential evidence for ribosomes, beginning with the discovery of ~20 nm diameter electron-dense particles rich in nucleic acids (*Palade, 1955*). Individual ribosomes (monosomes) are attached to the endoplasmic reticulum (ER) or free in the

cytosol, while mRNA-bound polysomes appear as beaded strings in EM images. To support the identity of ribosomes, EM analyses are often complemented by electron spectroscopic imaging (ESI) of ribosomal RNA (rRNA) phosphorus (*Koenig and Martin, 1996*; *Korn et al., 1983*), nucleic acid-binding dyes (*Glazer and Rye, 1992*; *Koenig and Martin, 1996*), ribosomal RNA antibodies (*Koenig et al., 2000*; *Lerner et al., 1981*), ribosomal protein antibodies (*Elkon et al., 1985*), or antibodies for proteins involved in translation, such as the eukaryotic translation initiation factor (*Tcherkezian et al., 2010*). These methods have been used to identify clustered ribosomes, often termed plaques, in squid neurons (*Martin et al., 1998*), goldfish Mauthner cell axons (*Koenig and Martin, 1996*), and mammalian peripheral axons (*Koenig et al., 2000*). The limitation of the aforementioned studies on ribosome localization is that they required tissues and/or cells to be fixed and therefore limited in their ability to address the mechanism of ribosome localization.

Until now, *in vivo* live imaging studies of functional ribosomes remain limited because labeling ribosomal proteins with a fluorescent protein presents challenges, compared to other organelles. First, as dozens of ribosomal proteins assemble into an RNA-rich macromolecule complex, there are limited sites for fluorescent protein tagging without interfering with ribosome function. Second, the expression level of ribosomal proteins is tightly controlled. Methodologies for live imaging of ribosomes largely rely on overexpression of ribosomal proteins tagged with fluorescence proteins (*Buxbaum et al., 2014*; *Rolls et al., 2002*), which can cause cellular defects (*Warner and McIntosh, 2009*). For example, overexpression of ribosomal subunit S29 causes apoptosis in primary cultured thymocytes of rats (*Khanna et al., 2000*). Moreover, overexpressed ribosomal proteins may not be incorporated into endogenous ribosomes, generating artifactual signals. Tagging ribosomal proteins at endogenous loci can solve this issue, but has been reported only in unicellular organisms. In bacteria, endogenous ribosomal protein S2 has been labeled with YFP (*Bakshi et al., 2012*). In yeasts, endogenous loci of ribosomal proteins have been tagged with GFP (*Huh et al., 2003*). For multicellular organisms, such as mice, knock-in of HA-tagged RPL22 (*Sanz et al., 2009*) or BAC transgenic expression of eGFP-tagged RPL10a (*Heiman et al., 2008*) has been used for ribosomal profiling of translational status. However, these reagents do not allow direct visualization of endogenous ribosomes in living tissues. A third challenge is the abundance of ribosomes in all tissues, potentially interfering with visualization of ribosomes in specific subcellular compartments, such as axons and dendrites. Thus, methods to visualize endogenous ribosomes in specific tissues in multicellular organisms have not been established and, as a result, few studies have addressed mechanisms controlling ribosome localization.

*Caenorhabditis elegans (C. elegans)* is ideally suited for visualizing protein and organelle localization *in vivo* because it is transparent and genetically tractable. Furthermore, candidate and forward genetic screens allow dissection of regulatory mechanisms. Here, we developed an approach using split GFP to visualize endogenous ribosomes in a tissue-specific manner. In *C. elegans* neurons, we found that ribosome localization is compartmentalized and displays dynamic changes during development and after axon injury. Loss of function of the Uncoordinated-16 (*unc-16*), encoding JNK-interacting protein 3 (JIP3), causes mislocalization of ribosomes. We further performed a forward genetic screen for mutants altering ribosome localization. Our analysis revealed roles of neuronal tubulins and the *mec-15* (*mec*hanosensory abnormality-15) encoding a F-box protein that functions as a regulator of tubulins, for proper ribosome localization. These findings support the utility of our approach for visualizing ribosomes in specific tissues and suggest that UNC-16/JIP3 and microtubules have specific roles in neuronal ribosome localization.

## Results

### Visualization of ribosomes using functional GFP-tagged ribosomal proteins

To visualize ribosomes *in vivo*, we selected ribosomal proteins whose N- and/or C-termini are on the surface of ribosomes, based on structural studies (*Ben-Shem et al., 2011*). We initially used standard multi-copy transgenes to overexpress GFP-tagged ribosomal proteins (see Materials and Methods). However, we noticed that such overexpression caused a variety of cellular defects, such as abnormal morphologies of axons and cell bodies (*Figure 1—figure supplement 1A*), and aggregates of ribosomal proteins in the nucleus (*Figure 1—figure supplement 2A*). Reasoning that tagged ribosomal

proteins should be functional and expressed at the levels comparable to endogenous ones, we then focused on ribosomal small subunit protein 18 (*rps-18*) or large subunit protein 29 (*rpl-29*), for which genetic null mutations were available. Homozygous *rps-18(ok3353)* mutants (*rps-18(0)*, **Figure 1A**) exhibited early larval arrest. On the other hand, homozygous *rpl-29(tm3555)* mutants (*rpl-29(0)*, **Figure 1—figure supplement 2B**) were superficially wild type, reflecting a possible non-essential role of *rpl-29* in ribosome function. We generated strains consisting of a single copy transgene expressing GFP-tagged *rps-18* or *rpl-29* fragment in the respective genetic null mutant background (Material and methods, **Supplementary file 1**). Our single-copy transgene *juSi83[Prps-18-GFP::rps-18]* (**Figure 1A**) rescued the *rps-18(0)* early larval arrest, but did not rescue sterility (**Figure 1—figure supplement 1C**), despite visible GFP expression in the germline (**Figure 1—figure supplement 1B**, gonad). Tagging with other fluorescent tags, such as mEOS, Dendra, and mini Singlet Oxygen Generator (miniSOG), also failed to fully rescue *rps-18(0)* phenotypes (data not shown), suggesting that large tags partly impaired RPS-18 function.

In the rescued somatic tissues, we observed GFP::RPS-18 in the cytosol, appearing as reticular structures (**Figure 1—figure supplement 1B**). RPL-29::GFP expressed from a single-copy transgene (*juSi123[Prpl-29-rpl-29::GFP])* in the *rpl-29(0)* background showed a similar pattern (**Figure 1—figure supplement 2C and D**). The similarity in localization of these large and small subunit proteins suggests that most of the signals represent assembled ribosomes as opposed to unassembled proteins or subunits. Importantly, the expression from these single-copy transgenes caused no detectable defects (data not shown) unlike the overexpression of *rps-18* or *rpl-29*.

We observed that the fluorescence intensity of P*rps-18*-GFP::RPS-18 was significantly lower in the *rps-18(+)/rps-18(+)* or *rps-18(+)/rps-18(0)* background than in homozygous *rps-18(0)/rps-18(0)* mutants (**Figure 1—figure supplement 1D and E**). Similarly, P*rpl-29*-GFP::rpl-29 signal intensity depended on the endogenous *rpl-29* locus (**Figure 1—figure supplement 2E**). These results suggest that negative feedback regulation eliminates excess ribosomal proteins, similar to that previously reported in bacteria (**Kaczanowska and Rydén-Aulin, 2007**). Therefore, the null mutant background is necessary to obtain sufficiently bright signals for visualization of GFP-tagged ribosomal proteins.

## Tissue-specific Ribosome Imaging Based On Split GFP (RIBOS)

The ubiquitous expression of GFP::RPS-18 in all tissues precluded analysis of ribosome localization in individual cells or cellular compartments. Moreover, tagging with relatively large fluorescent proteins, such as GFP, might interfere with the function of ribosomal proteins. To circumvent these challenges, we devised a strategy using self-assembling split GFP; when the first ten β-strands (GFP1-10) and the last β-strand (GFP11) coexist, they irreversibly assemble and produce green fluorescence (**Cabantous et al., 2005**). We reasoned that GFP11-tagged endogenous proteins could be visualized by expressing GFP1-10 under the control of a tissue-specific promoter (**Figure 1B**). Importantly, we found that a single-copy expression of GFP11::RPS-18 (*juSi94[GFP11::rps-18]*) completely rescued larval arrest and sterility of *rps-18(0)* (**Figure 1—figure supplement 1C**). This is probably because GFP11 is much smaller (16 amino acids), compared to GFP (237 amino acids), supporting our interpretation for partial impaired function of the GFP::RPS-18. We then expressed GFP1-10 in specific tissues, such as epidermis, muscles, or neurons to visualize GFP11::RPS-18 expression in the *juSi94; rps-18(0)* background (**Figure 1B** and see below). Since the binding between GFP11 and GFP1-10 is irreversible, it is possible that the reconstituted GFP::RPS-18 is not fully functional in contexts where ribosome activity is locally required or sensitive to dosage. Nonetheless, the subcellular fluorescence pattern revealed by this split-GFP transgenic approach was similar to that from the single-copy expression of GFP::RPS-18 or of RPL-29::GFP. Here, we call this approach <u>R</u>ibosome <u>I</u>maging <u>B</u>ased <u>O</u>n <u>S</u>plit GFP (RIBOS). We denote tissue-specific RIBOS, namely, *juSi94[GFP11::rps-18]; rps-18(0); juEx or juIs[Pgene-GFP1-10]*, where P*gene* is a tissue-specific promoter, as 'P*gene*-RIBOS' for simplicity.

We first evaluated RIBOS signals in non-neuronal tissues. In adult epidermis-specific P*col-19*-RIBOS, we observed signals in the epidermal cytosol excluded from reticular structures (**Figure 1C**, Epidermis). Although RPS-18 is expressed in the whole body, fluorescence was not observed in any other tissues, such as intestine (**Figure 1C**, Intestine), suggesting that GFPP11::RPS-18 did not have fluorescence. Worms expressing P*col-19*-GFP1-10 transgene alone also did not show fluorescent signals compared to P*col-19*-RIBOS (**Figure 1C** and Materials and Methods). To address whether GFP11::RPS-18 was incorporated into endogenous ribosomes, we measured Fluorescence Recovery

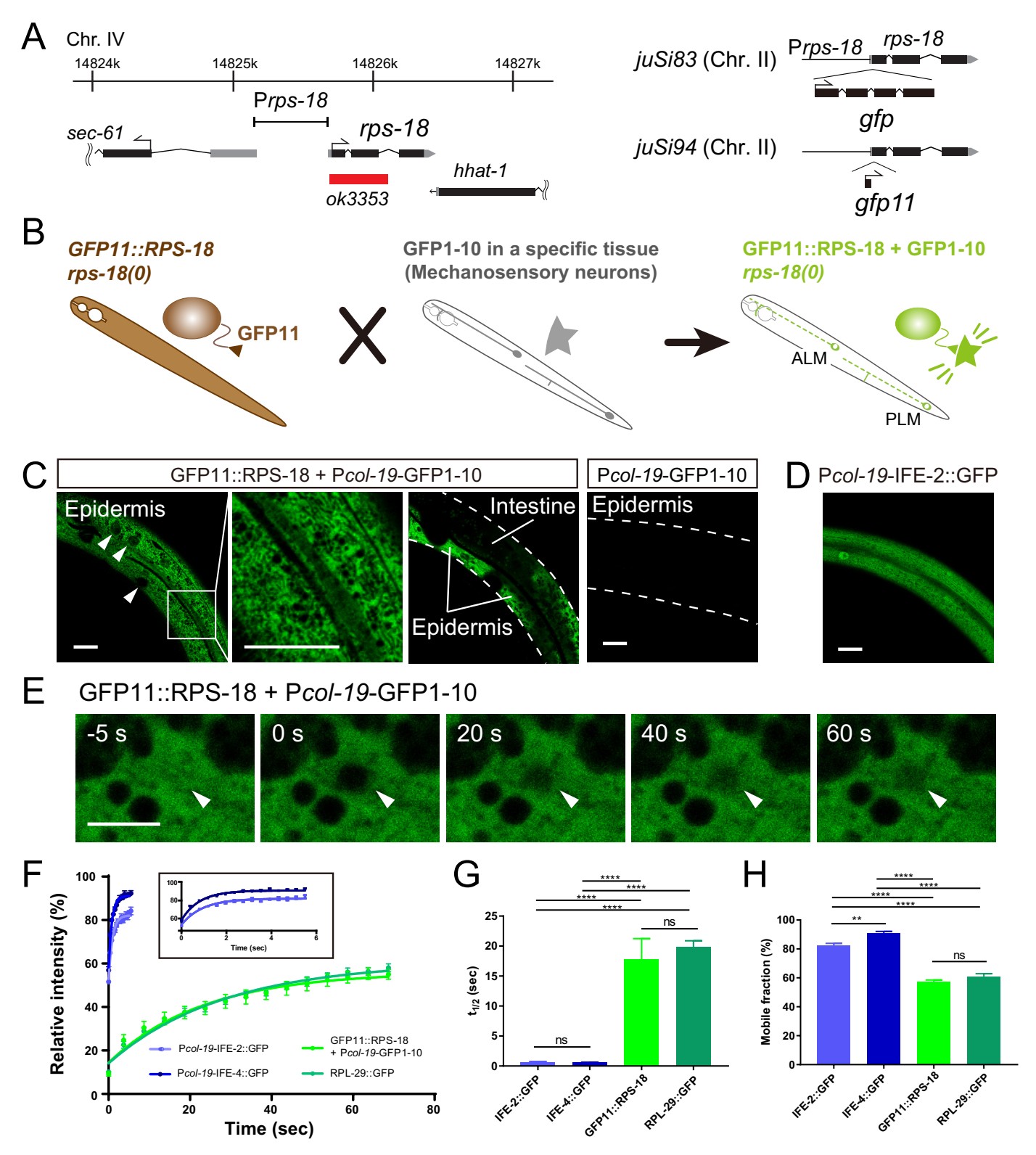

**Figure 1.** Ribosome Imaging Based On split GFP (RIBOS). (**A**) Schematics of endogenous *rps-18* locus on chromosome IV and single-copy transgenes on chromosome II. Black box: exon, Grey box: untranslated region, Red box: deletion. (**B**) Schematic of RIBOS for mechanosensory neurons. The *rps-18* fused to the smaller part of split GFP (GFP11) is replaced to the endogenous *rps-18* (*juSi94[GFP11::rps-18]*; *rps-18(0)*, left). The larger part of split GFP (GFP1-10) is overexpressed using a high-copy transgene in a targeted tissue (middle). After crossing these strains, GFP1-10 binds to the GFP11::RPS-18

*Figure 1 continued on next page*

*Figure 1 continued*

in the targeted tissue and visualize RPS-18 (right). (**C**) Epidermis-specific P*col-19*-RIBOS (*juSi94[GFP11::rps-18]; rps-18(0); juEx5375[Pcol-19-GFP1-10]*). The RIBOS signals were excluded from the nuclei (Left panel, arrowheads) and reticular structures (magnified image). Although GFP11::RPS-18 is expressed in the intestine, it had no signals. The negative control only expressed GFP1-10 (*juEx5375*) in the epidermis. Scale bars: 20 µm. (**D**) Diffuse IFE-2 expression visualized by *juEx5809[Pcol-19-IFE-2::GFP]*. (**E**) Representative images of the FRAP experiment using P*col-19*-RIBOS. The fluorophore was bleached in the area (arrowheads) at 0 s. Scale bars: 2 µm. (**F**) Fluorescent recovery after photobleaching was plotted for *juEx5809[Pcol-19-IFE-2:: GFP]*, *juEx5811[Pcol-19-IFE-4::GFP]*, P*col-19*-RIBOS, and *juSi123[RPL-29::GFP]; rpl-29(0)*. The line represents the one-phase fit to an exponential function for each plot. The inset shows the magnified graph for IFE-2::GFP and IFE-4::GFP. Error bars indicate S.E.M. (**G and H**) $t_{1/2}$ and mobile fraction calculated from (**F**). n= 5 or 6. Error bar indicates S.E.M., Statistics: One-way ANOVA, ns: $p>0.05$, $p**<0.01$, $p****<0.0001$.
DOI: https://doi.org/10.7554/eLife.26376.002
The following figure supplements are available for figure 1:

**Figure supplement 1.** GFP::RPS-18 visualizes ribosomes in the whole body.
DOI: https://doi.org/10.7554/eLife.26376.003
**Figure supplement 2.** Ribosome visualization with RPL-29::GFP.
DOI: https://doi.org/10.7554/eLife.26376.004
**Figure supplement 3.** RIBOS visualizes ribosomes in muscle arms.
DOI: https://doi.org/10.7554/eLife.26376.005

After Photobleaching (FRAP) of P*col-19*-RIBOS (*Figure 1E*). If GFP11::RPS-18 is incorporated into a large organelle or complex, such as the ribosome, the half time of fluorescence recovery ($t_{1/2}$) should be larger, and the mobile fraction should be smaller than those of a free cytosolic protein (*Lippincott-Schwartz et al., 2001*). We used GFP-tagged Initiation Factor 4E (eIF4E) homologs, IFE-2::GFP (*Song et al., 2010*) and IFE-4::GFP (*Dinkova et al., 2005*), as controls because their diffuse distribution implied lack of association to specific protein complex or organelle (*Figure 1D*). Photobleached GFP11::RPS-18 showed a longer recovery time and smaller mobile fraction than IFE-2::GFP and IFE-4::GFP (*Figure 1F–1H*). Furthermore, the recovery curve, $t_{1/2}$, and mobile fraction of GFP11::RPS-18 were comparable to that of the ribosomal large subunit, RPL-29::GFP (*Figure 1F–1H*), suggesting that fluorescence recovery was not affected by splitting GFP, and that RPS-18 and RPL-29 were incorporated into a complex with a similar mobility, likely the ribosome.

Body wall muscle-specific P*myo-3*-RIBOS revealed that besides strong somatic signals, fluorescence was observed in the muscle arm (*Figure 1—figure supplement 3A*), an elaborate postsynaptic structure extending to the nerve cord to receive inputs from presynaptic neurons (*Dixon and Roy, 2005*). Such RIBOS signals were consistent with the distribution of ribosomes in muscle arms as found in electron micrographs (*Figure 1—figure supplement 3B*). These data suggest that RIBOS is suitable for visualizing ribosome localization in a specific tissue. Below, we refer to RIBOS signals as ribosomes.

## Developmental changes of ribosome localization in neurons

We next examined ribosome localization in neurons. Pan-neuronal P*rgef-1*-RIBOS showed that ribosomes were mostly restricted to the soma (*Figure 2A–C*). In contrast to the restricted signals of ribosomes, initiation factors IFE-2::GFP and IFE-4::GFP were diffuse in cell bodies and nerve processes in the ventral nerve cord (*Figure 2C*). We also examined the neuronal localization of two putative components of RNA granules, which have been suggested to contain ribosomes (*Elvira et al., 2006*). STAU-1 (*LeGendre et al., 2013*), an ortholog of Staufen, formed two foci in the soma; AIN-1 (ALG-1-interacting protein-1), an ortholog of GW182 (*Ding et al., 2005*), formed puncta in the soma and along the neuronal processes (*Figure 2C*). Neither the initiation factors nor the RNA granule components showed localization similar to ribosomes, implying that protein translation-related components might be differentially regulated in their localization.

To examine ribosomes in axonal or dendritic processes of specific neurons, we used the neuronal subtype-specific promoters, as the abundance of ribosomal signals using a pan-neuronal promoter hindered data analysis. P*unc-25*-RIBOS labeled ribosomes in the GABAergic dorsal D-type (DD) and ventral D-type (VD) motor neurons. In the first larval (L1) stage, we found that ribosomes colocalized with presynaptic markers, RAB-3, in the ventral cord, corresponding to the presynaptic terminals of DD motor neurons (*Figure 3A*). At late L1 to L2 stage DD neurons remodel their pre-synaptic specializations from the ventral cord to the dorsal cord, in a phenomenon known as DD remodeling

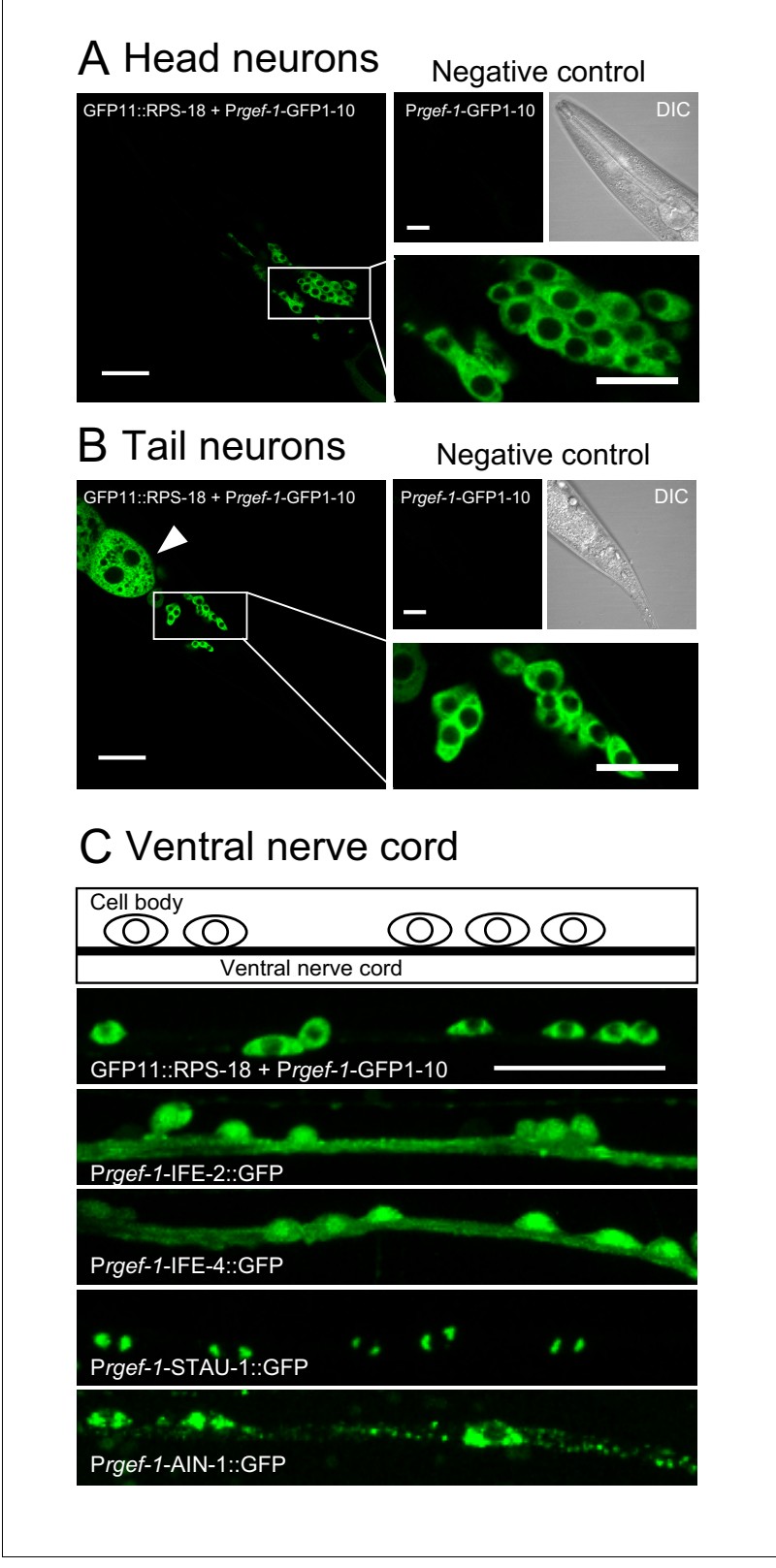

**Figure 2.** Ribosomes are mostly restricted in the neuronal soma. (**A and B**) Pan-neuronal P*rgef-1*-RIBOS visualizing ribosomes (*juSi94[GFP11::rps-18]; rps-18(0) juIs409[Prgef-1-GFP1-10]*) in head neurons (**A**) and tail neurons (**B**). Negative controls only expressed GFP1-10 (*juIs409[Prgef-1-GFP1-10]*). An arrowhead in (**B**) indicates ribosomes in the intestine due to the leak expression of GFP1-10. Scale bars: 20 μm; 10 μm in the magnified images. (**C**)

*Figure 2 continued on next page*

*Figure 2 continued*

Fluorescent images and a schematic of motor neurons in the ventral nerve cord. P*rgef-1*-RIBOS is shown in the upper left panel. Initiation factors (IFE-2::GFP and IFE-4::GFP) and putative components of RNA granules (AIN-1::GFP and STAU-1::GFP) were expressed under the control of the *rgef-1* promoter using extrachromosomal arrays (see *Supplementary file 2* for the transgenes). Scale bar: 20 μm.
DOI: https://doi.org/10.7554/eLife.26376.006

(*White et al., 1978*). Concurrent with DD remodeling, the VD neurons form presynaptic specializations along the ventral cord. From L2 to older larvae, we observed P*unc-25*-RIBOS signals in the ventral cord, which were excluded from VD presynaptic terminals (*Figure 3A*). DD neuron-specific P*flp-13*-RIBOS showed similar localization (*Figure 3B*), suggesting that ribosome localization was changed from presynaptic to postsynaptic compartments in the DD neurons. Consistent with RIBOS, reconstruction of serial EM sections showed that the number of ribosomes in DD axonal profiles was higher at the L1 stage than the adult stage (*Figure 3C*). In the adult presynaptic terminals, the evidence of ribosome localization has been limited (*Akins et al., 2009*). Here, we observed ribosomes in the presynaptic terminals of adult VD neurons in serial EM sections (*Figure 3C*). Using electron tomography, we further confirmed the existence of ribosomes in the presynaptic terminals of those neurons (*Figure 3D and E*), based on the following criteria: ribosomes in neurons had spherical shapes with 20 nm diameter, and resembled ribosomes in muscles (*Figure 3E–H*). We were unable to detect RIBOS signals in the ventral cord colocalized with presynaptic markers at the adult stage, likely due to the number of ribosomes being below that of light microscopy detection threshold.

To examine a different type of neurons, we observed the developmental changes of ribosome distribution in mechanosensory neurons. We used P*mec-4*-RIBOS labeling Anterior and Posterior Lateral Microtubule cells (ALM and PLM, respectively), in which local translation (*Yan et al., 2009*) and axonal ribosome localization (*Topalidou et al., 2012*) have been reported. In these neurons, we observed ribosomes in the soma and the proximal axon (~20 μm from the soma, *Figure 4A*). Ribosomes also formed puncta along the axon and puncta density decreased from L1 to adult stage in PLM neurons (*Figure 4A–4C*). At the L1 stage, but not at the adult stage, these puncta were associated with the junction of the synaptic branch of PLM neurons (*Figure 4B and D*). These data indicate that ribosomal localization changes during the neuronal development.

## Ribosomes accumulate at the tip of the proximal axon after axon injury

In *C. elegans* mechanosensory neurons, local translation in axons is induced after axon injury (*Yan et al., 2009*). Therefore, we next asked if neuronal ribosome localization in adult animals was altered after axon injury. We severed the proximal axonal process of PLM mechanosensory neurons by laser axotomy (*Wu et al., 2007*). Around 6 hr after axotomy, microtubules are reorganized, leading to the formation of a growth cone, and severed axons start regrowing around the same time, resulting in average regrowth about ~100 μm after 24 hr (*Chen et al., 2011*; *Ghosh-Roy et al., 2012*). Axon regrowth partly depended on new protein synthesis because treatment with cycloheximide, which blocks protein synthesis, inhibited this regrowth (*Figure 5—figure supplement 1*). One hour after axon injury, ribosome accumulation began at the injured site of the proximal axon and appeared to plateau after three hours (*Figure 5A and B*). We also observed weak RIBOS signals in the regrowing tip (*Figure 5A*). Negative controls involving no wounding or laser wounding without severing the axon did not induce ribosome accumulation (*Figure 5B*). Consistent with RIBOS, reconstruction of EM serial sections showed accumulation of electron-dense monosomes at the cutting site six hours after axotomy (*Figure 5D and E*). We also observed mitochondria around the cutting site in the EM images (*Figure 5D and E*). The uninjured axon contained no ribosomes or mitochondria (*Figure 5C*). This is consistent with the previous observation showing few ribosomes in mechanosensory axons (*Topalidou et al., 2012*). These results suggest that axonal ribosomes at the adult stage can alter their localization after axon injury.

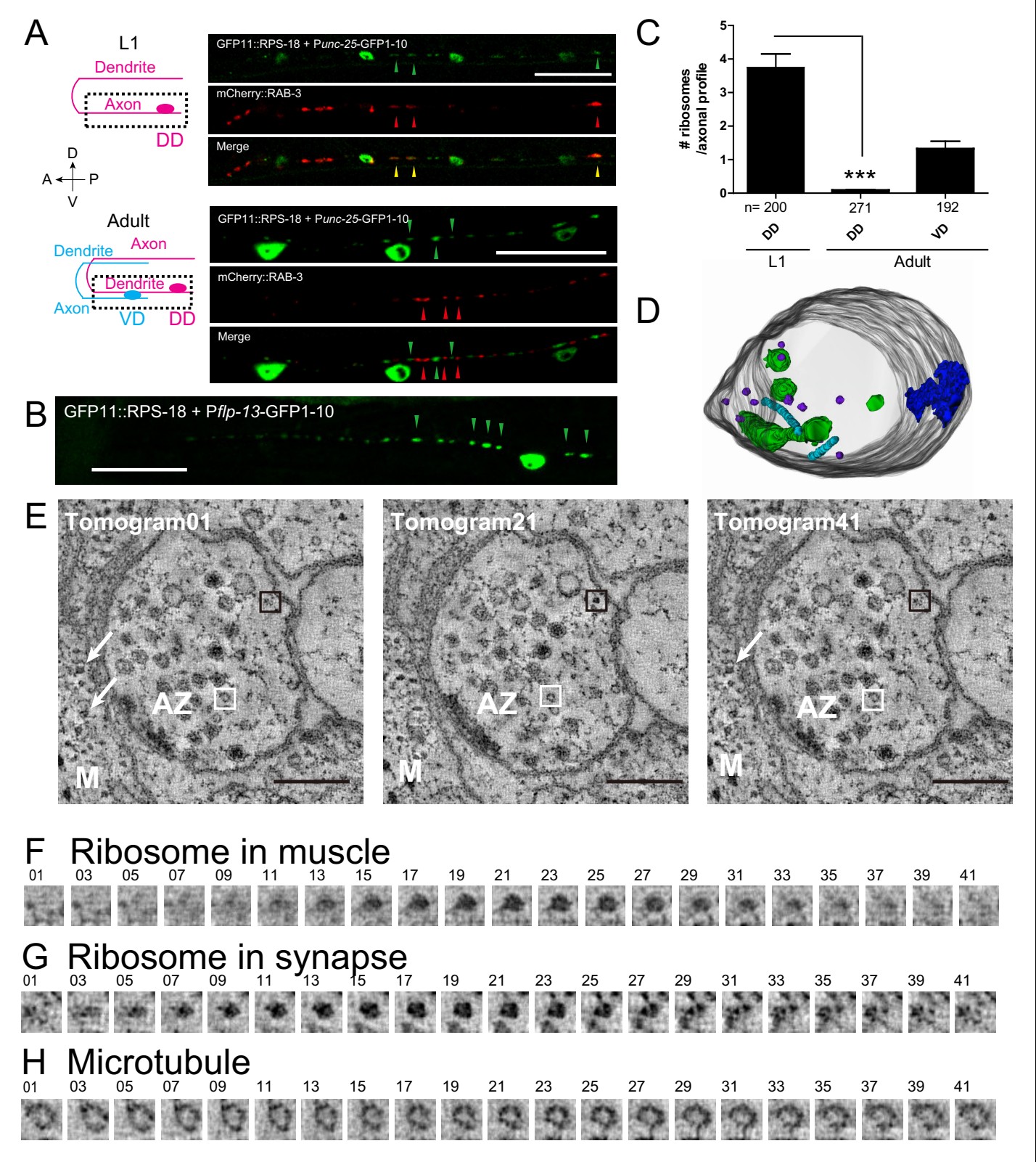

**Figure 3.** Ribosomes localize to the synaptic compartment in motor neurons. (**A**) Schematic illustrates GABAergic motor neurons at the L1 or adult stage, in which dotted squares indicate imaged regions. As VD neurons are born, DD neurons change the innervation. GABAergic motor neuron-specific P*unc-25*-RIBOS (green arrowheads) and a presynaptic marker P*unc-25*-mCherry-RAB-3 (red arrowheads) were colocalized at the L1 stage (yellow arrowheads) but not at the adult stage. Scale bars: 20 µm. (**B**) DD-neuron specific P*flp-13*-RIBOS showed punctate signals in the ventral cord (green

*Figure 3 continued on next page*

*Figure 3 continued*

arrowheads). Scale bar: 20 μm. (C) The number of ribosomes were counted in the axonal profiles of serially reconstructed electron micrographs for GABAergic motor neurons. Numbers of axonal profiles are indicated. Statistics: Student's t-test, ***p<0.001. (D) A representative presynaptic volume reconstructed from EM tomograms of the adult motor neurons in the ventral cord, showing that presynaptic terminals contain ribosomes. Purple objects: ribosomes, blue: active zones, light blue: microtubules, green: endosomes. (E) Representative electron tomograms of presynaptic terminal of a motor neuron in the ventral nerve cord of an adult animal with 12 nm intervals. The active zone is surrounded by synaptic vesicles and dense core vesicles. White arrowheads indicate ribosomes in muscles (M). White boxes mark microtubules, which have a similar diameter as a ribosome; black boxes ribosomes. (F–H) Electron tomograms with 1.2 nm intervals for a ribosome in muscle (F), and a ribosome (G) and a microtubule (H) in the presynaptic terminal, showing that a microtubule and a ribosome have a similar diameter, but microtubule is continuous in z-direction unlike ribosome.

DOI: https://doi.org/10.7554/eLife.26376.007

## The JIP3 protein UNC-16 affects ribosome distribution in presynaptic terminals

To investigate mechanisms underlying neuronal ribosome distribution, we examined mutants of select candidate genes for ribosome distribution using mechanosensory neuron-specific P*mec-4*-RIBOS. Since ribosomes are mostly observed in the neuronal soma, we focused on proteins which

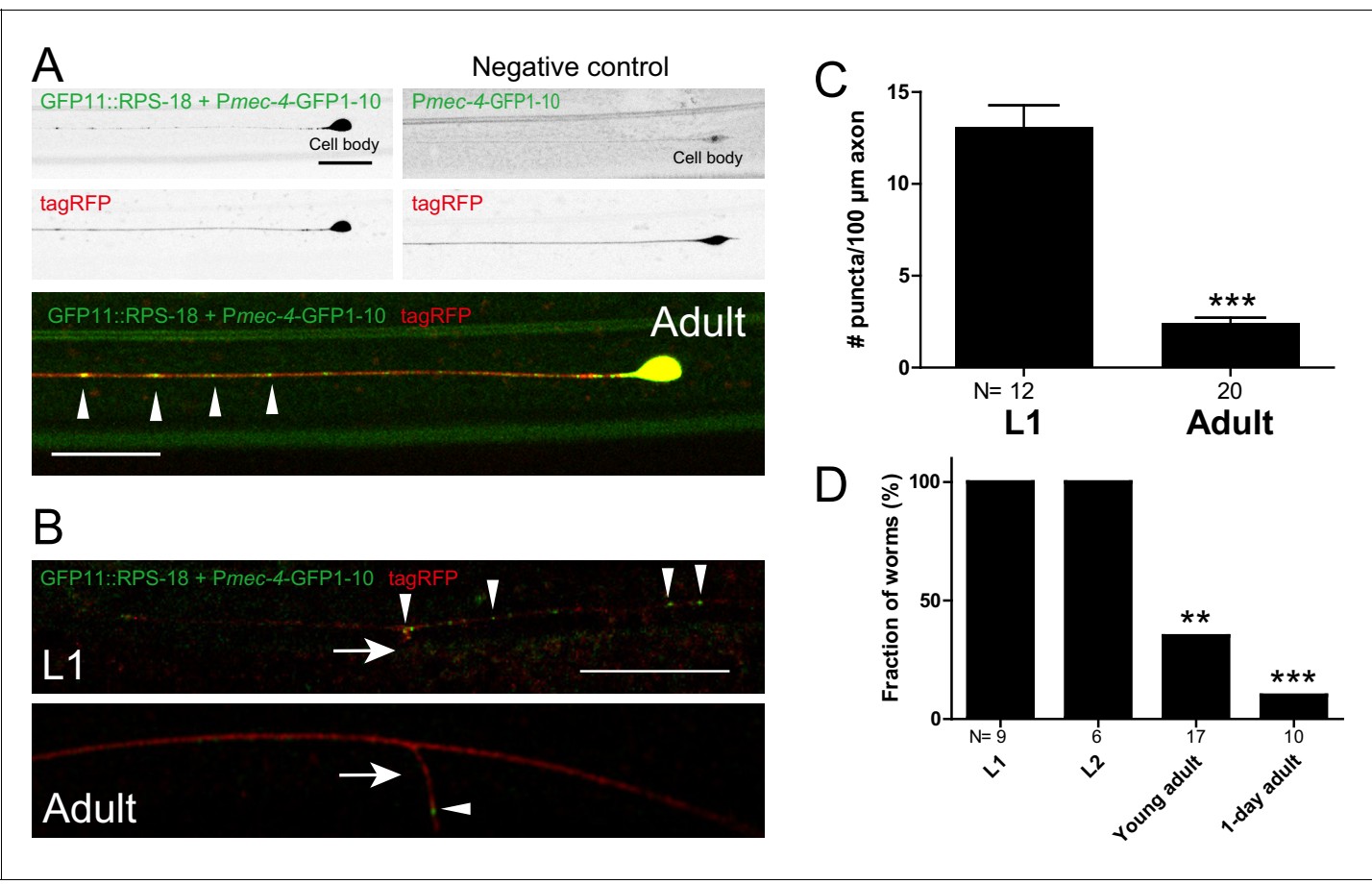

**Figure 4.** Ribosomes form puncta along the sensory neuron axons. Mechanosensory neuron-specific P*mec-4*-RIBOS (*juSi94[GFP11::rps-18] juIs438 [Pmec-4-GFP1−10+Pmec-4-tagRFP]; rps-18(0)*). (A) ALM neurons at the adult stage. Negative control strain expressing only GFP1-10 had weak signals in the nucleus. Ribosomes form puncta along the ALM axon (arrowheads). Scale bars: 20 μm. (B) Comparison of RIBOS signals between L1 and adult stages around the synaptic branch of PLM neurons. Arrows and arrowheads indicate synaptic branches and RIBOS signals, respectively. Scale bar: 20 μm. (C) Number of puncta per 100 μm axon around the PLM branch. Statistics: Student's t-test, ***p<0.001. N: number of worms. (D) Fraction of worms with ribosomes associated with PLM branch point at indicated developmental stages. Statistics: Fisher's exact test, compared to the L1 stage. **p<0.01, ***p<0.001. N: number of worms.

DOI: https://doi.org/10.7554/eLife.26376.008

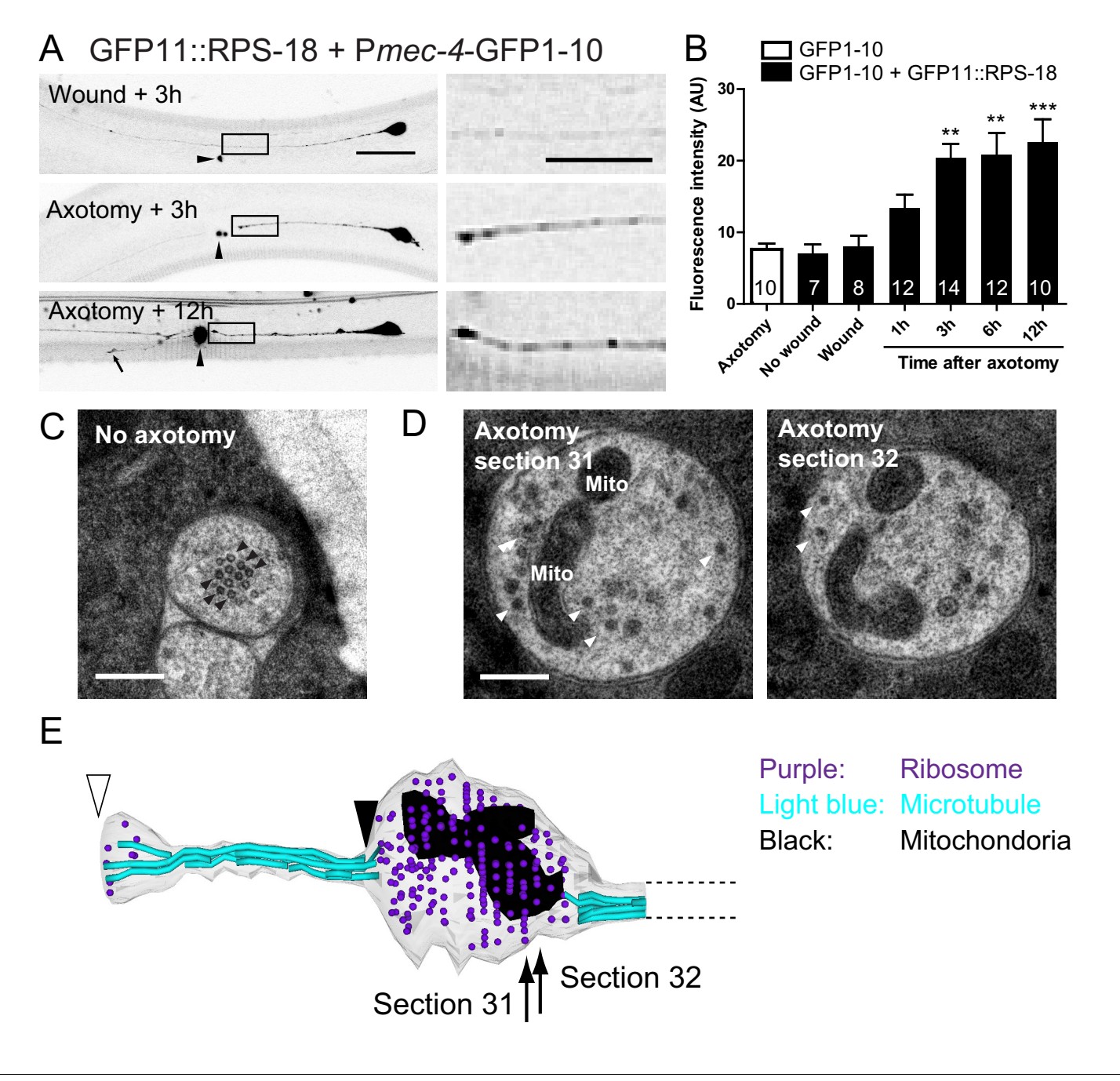

**Figure 5.** Ribosomes accumulate at the tip of proximal axons after axon injury in mechanosensory neurons. (**A**) P*mec-4*-RIBOS three or twelve hours after axon injury or wounding. For negative control, worms were wounded by laser without damaging the axon. Right panels are magnified images of the black square regions. Arrowheads indicate laser illumination site. An arrow indicates the tip of the regrowing axon. Scale bar: 20 μm; 10 μm in the magnified image. (**B**) Quantification of fluorescent intensity at the proximal axons indicated by boxes in (**A**). Numbers of worms were shown in the bars. Statistics: One-way ANOVA, **$p < 0.01$, ***$p < 0.001$, compared to the wounded condition. (**C**) Electron micrograph of PLM axon without axon injury. The uninjured PLM on the right side was used as a control from the same worm as in (**D**). Black arrowheads: microtubules. Scale bar: 200 nm. (**D and E**) The tip of the PLM axon on the left side six hours after axon injury. (**D**) Two serial electron micrographs corresponding to two sections in (**E**). White arrowheads: monosome-like electron dense particles; Mito: mitochondria. Scale bar: 200 nm. (**E**) Serial reconstruction EM, corresponding to forty-two sections with 50 nm thickness (2.1 μm in length). Black arrowhead indicates the presumable cutting site and white arrowhead the regrowing tip. Ribosome (purple dots) and mitochondria (black) accumulated around the cutting site, which lacks microtubules (light-blue lines).

DOI: https://doi.org/10.7554/eLife.26376.009

*Figure 5 continued on next page*

*Figure 5 continued*

The following figure supplement is available for figure 5:

**Figure supplement 1.** Translation inhibitor prevents axon regrowth.

DOI: https://doi.org/10.7554/eLife.26376.010

might prevent ribosomes entering neuronal processes. In mammals, large molecules are excluded from the axons of mature neurons by the axon initial segment (*Song et al., 2009*). Knockdown of a key regulator of the axon initial segment, ankyrin-G, allows somatodendritic components to move into the axon (*Hedstrom et al., 2008*; *Song et al., 2009*). β-spectrin is another component of the axon initial segment and binds ankyrin-G (*Komada and Soriano, 2002*). We found normal ribosome distribution in the soma and axon in loss-of-function mutants of *C. elegans* orthologs for ankyrin (*unc-44*) and its putative binding partner spectrin (*unc-70*) (*Figure 6*), indicating that these genes may not play a role in restricting ribosomes in the soma. The JNK-interacting protein 3 (JIP3) ortholog, UNC-16, has been shown to prevent endosomal organelles from exiting out of the soma (*Edwards et al., 2013*). In PLM neurons of *unc-16* mutants, ribosomes were decreased in the proximal axon and increased in the presynaptic terminals without affecting somatic distribution (*Figure 6*), suggesting that UNC-16 acted as a gatekeeper for ribosomes in mechanosensory neurons.

## A forward genetic screen identified roles for tubulins in ribosome distribution

Our identification of *unc-16* by the candidate approach gave us the confidence to identify additional ribosomal regulators by unbiased forward genetic screens. Thus, we performed a genetic screen using P*mec-4*-RIBOS and found eight mutants displaying aberrant ribosomal distribution (see Materials and methods). Ribosomes in these mutants accumulated in the cell cortex and around the nucleus, compared to the relatively diffused distribution in the wild type, and increased in the proximal axon with partial penetrance (*Figure 7A and B*). In addition to these ribosome defects, all mutants had misshapen mechanosensory neuron cell bodies, and insensitivity to gentle touch. Based on complementation tests and sequencing, we found that all mutations affect tubulin or tubulin regulatory genes, previously isolated as mechanosensory abnormality (*mec*) genes (*Table 1*). The ALM and PLM mechanosensory neurons are filled with unique 15-protofilament microtubules (*Chalfie and Sulston, 1981*), which consist of α- and β-tubulins encoded by *mec-12* (*Fukushige et al., 1999*) and *mec-7* genes (*Savage et al., 1989*), respectively. Four recessive mutations (*ju1286*, *ju1287*, *ju1294*, and *ju1298*) and a dominant mutation *ju1297* affecting ribosome localization were found to be substitution mutations in *mec-7* (*Table 1* and *Figure 7—figure supplement 1A*). All four recessive mutations are located in the exchangeable GDP/GTP-binding site of the dimer-dimer interface of α- and β-tubulins (*Figure 7—figure supplement 1D*, magenta). In contrast, the dominant mutation is located in the non-exchangeable GTP-binding site of the monomer-monomer interface (*Figure 7—figure supplement 1D*, cyan). One dominant allele, *ju1295*, was mapped to *mec-12* (*Figure 7—figure supplement 1B*) and caused a Glu to Ser substitution on the exchangeable GDP/GTP-binding site in the dimer-dimer interface (*Figure 7—figure supplement 1D*, light green). A deletion allele *mec-7(u443)* and a recessive loss-of-function allele of *mec-12(e1607)* (*Figure 7—figure supplement 1D*, light blue) exhibited comparable ribosome distribution defects (*Figure 7D*), suggesting that the identified mutations were causative. Interestingly, dominant alleles, *mec-7(ju1297)* and *mec-12 (ju1295)* showed lower penetrance of ribosome defects than recessive alleles of *mec-7* only in the axons (*Figure 7B*). This result suggests that axonal and somatic ribosomes might be differentially regulated. Two recessive mutations, *ju1288* and *ju1289*, were mapped to the E3 ubiquitin ligase homolog, *mec-15* (*Figure 7—figure supplement 1C*), which regulates expression of *mec-7* and *mec-12* genes (*Bounoutas et al., 2009b*; *Gu et al., 1996*). We then tested a mutant of the α-tubulin acetyltransferase, *mec-17* (*Akella et al., 2010*), and found that it showed similar ribosome distribution defects (*Figure 7D*). These results suggest that tubulin genes and their regulation are important for ribosome distribution.

In addition to tubulins, the *mec* class includes genes encoding mechanosensory channels (*mec-4*, *mec-6*, and *mec-10*), extracellular matrix (*mec-1*, *mec-5*, and *mec-9*), and an adaptor protein (*mec-2*) as schematized in *Figure 7C*. In contrast to the tubulin genes, *mec-4* and *mec-2* mutants did not

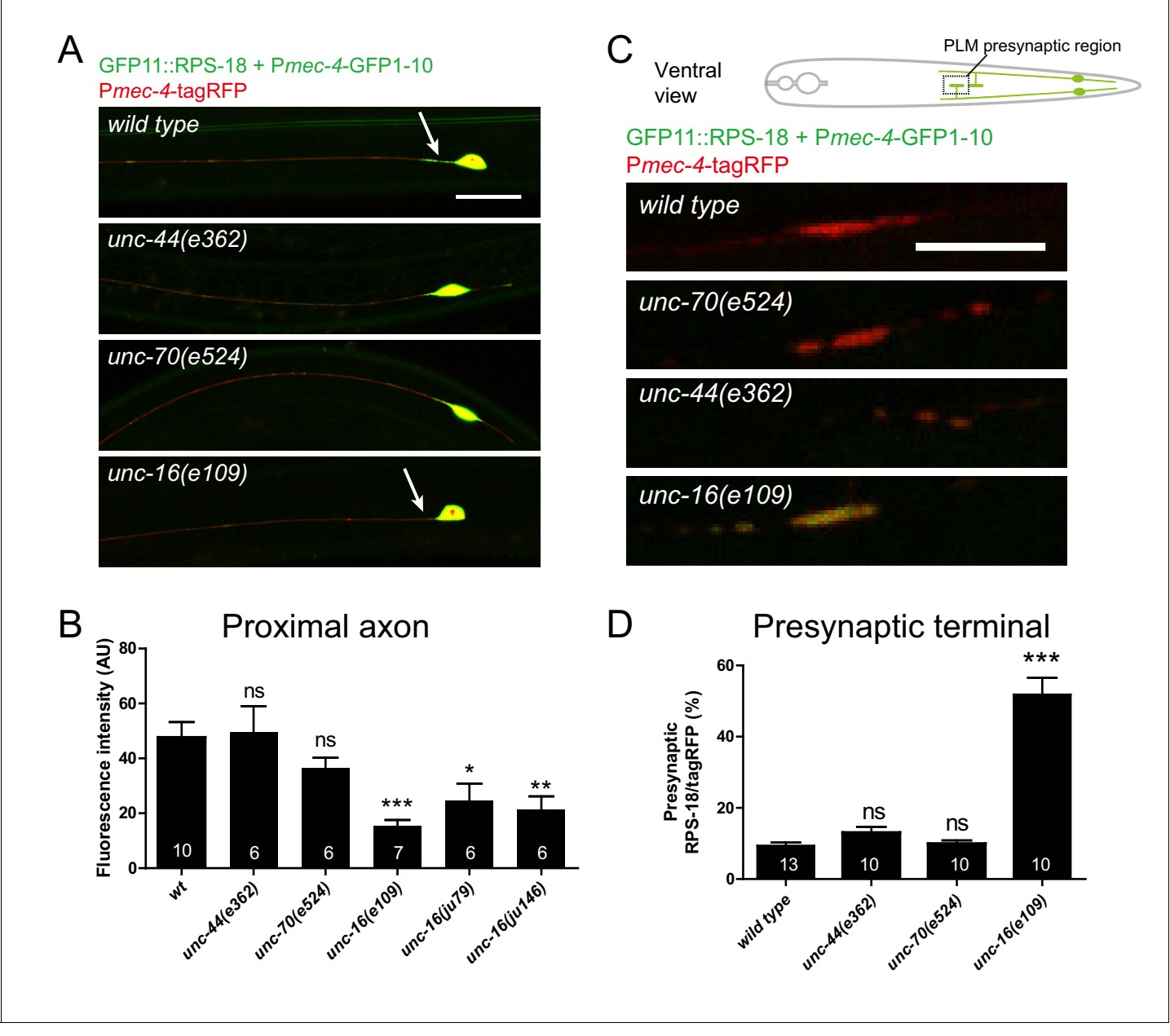

**Figure 6.** Ribosome distribution is altered in *unc-16* mutants. Mechanosensory neuron-specific Pmec-4-RIBOS with free tagRFP. (**A**) Representative images of ALM neurons in the indicated genetic backgrounds. *unc-16* mutants showed reduced ribosomes in the proximal axon (arrows). Scale bar: 20 µm. (**B**) RIBOS fluorescence intensity in the proximal axon (20 µm from the soma). Numbers of worms are shown in the bars. Statistics: One-way ANOVA, ns: p>0.05, *p<0.05, **p<0.01, ***p<0.001. (**C**) PLM presynaptic terminals in the ventral cord, highlighted in the schematic. *unc-16* mutants showed more ribosomes. (**D**) Ratio of RIBOS to tagRFP in the presynaptic terminals. Numbers of worms were shown in the bars. Statistics: One-way ANOVA, ns: p>0.05, ***p<0.001.

DOI: https://doi.org/10.7554/eLife.26376.011

show ribosome distribution defects (**Figure 7D**). Furthermore, *mec-12(e1605)* mutants, which are mechanosensory defective but have intact microtubules (**Bounoutas et al., 2009a**), did not show ribosome distribution defects (**Figure 7D**). Together, these results suggest that ribosome distribution defects are not a direct consequence of mechanosensory defects.

To distinguish between roles of filamentous microtubules and globular tubulins in ribosome distribution, we treated worms with colchicine, which disrupts filamentous microtubules in

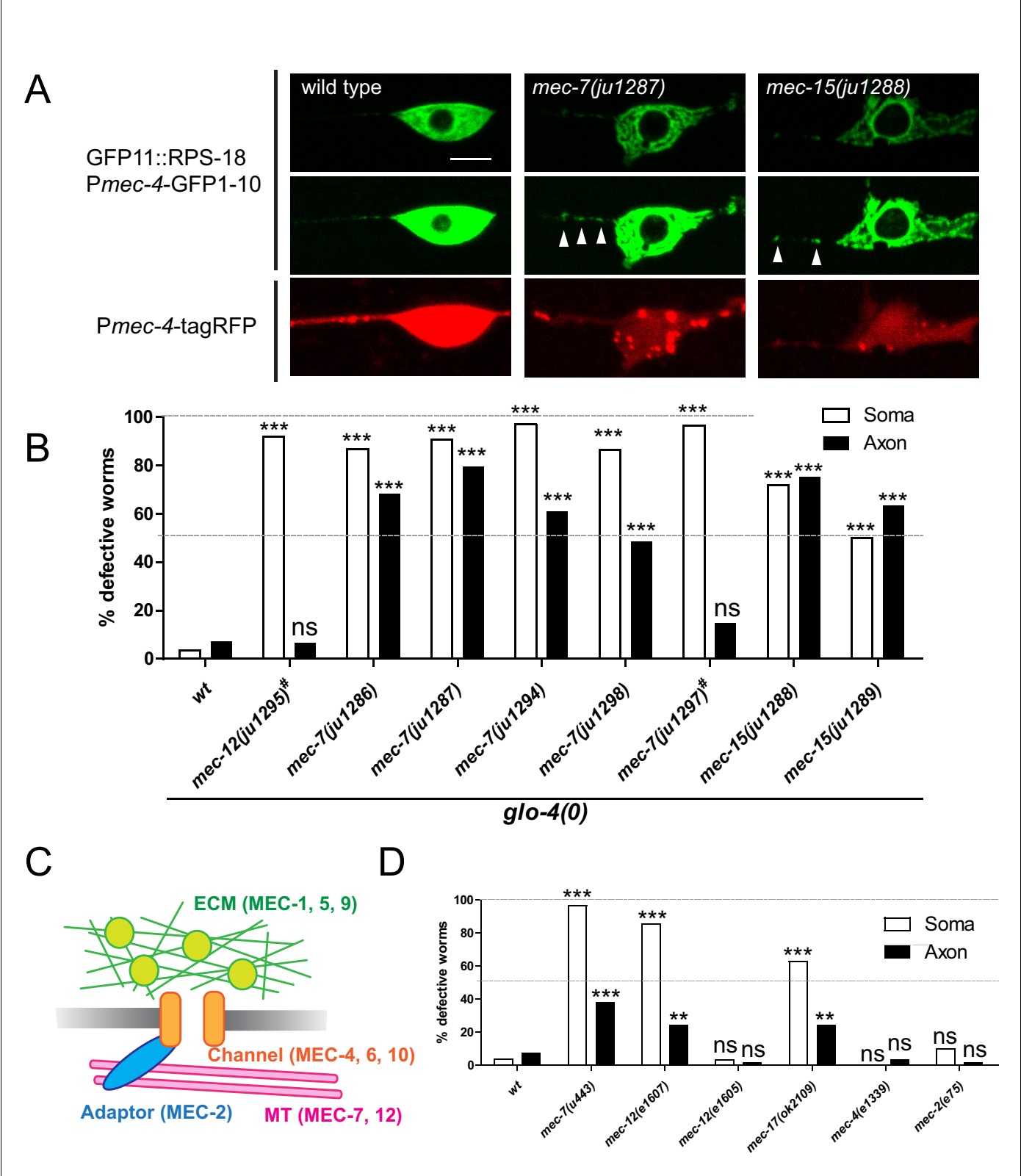

**Figure 7.** Ribosome distribution defects in tubulin mutants. (**A**) Ribosome distribution in the ALM soma and proximal axon using P*mec-4*-RIBOS in the indicated backgrounds. Brightness was adjusted in the middle panels to show the puncta in the proximal axons in the *mec-7* and *mec-15* mutants (white arrowheads). TagRFP intensity was decreased in the *mec-7* and *mec-15* mutants (bottom panels), consistent with a previous report (***Bounoutas et al., 2011***). Scale bar: 5 μm. (**B and D**) The fraction of worms showing the ribosome distribution defects in the soma or in the axon of

*Figure 7 continued on next page*

*Figure 7 continued*

ALM neurons in the indicated genetic backgrounds. # indicates dominant mutants. 50–70 worms were analyzed. Statistics: Fisher's exact test, compared to the wild type control, ns: $p>0.05$, $**p<0.01$, $***p<0.001$. (C) Schematic of mechanotransduction components in *C. elegans*. Representative proteins of each component are shown in brackets. ECM: extracellular matrix; MT: microtubules.

DOI: https://doi.org/10.7554/eLife.26376.012

The following figure supplements are available for figure 7:

**Figure supplement 1.** Gene and protein structures of ribosome defective mutants.

DOI: https://doi.org/10.7554/eLife.26376.013

**Figure supplement 2.** Microtubules are important for maintenance of ribosome distribution.

DOI: https://doi.org/10.7554/eLife.26376.014

mechanosensory neurons (*Chalfie and Thomson, 1982*). Chronic treatment from eggs induced ribosome distribution defects similar to *mec-7* or *mec-12* mutants, suggesting that filamentous microtubules, but not globular tubulins, were important (*Figure 7—figure supplement 2A*, Egg). Acute treatment at the L4 or young adult stage altered ribosome somatic distribution (*Figure 7—figure supplement 2A*). Also, somatic defects emerged at the adult stage in *mec-7* mutants (*Figure 7—figure supplement 2B*). These results suggest that microtubules are important for the maintenance of ribosome distribution in the soma. In contrast, axonal defects were not induced by acute colchicine treatment (*Figure 7—figure supplement 2A*) and emerged at the larval stage in the *mec-7* mutants (*Figure 7—figure supplement 2B*), suggesting that microtubules might be important for the generation of a barrier in the proximal axon.

## Discussion

Here, we established an approach using split GFP to visualize endogenous ribosomes in specific cells in live animals, and demonstrated the utility of such reporters in genetic screens. Fluorescent signals representing ribosomes were punctate in the neuronal processes in motor and mechanosensory neurons, and change during development or after axon injury. Furthermore, our analyses revealed roles for JIP3/*unc-16* and microtubules in regulating ribosome distribution in neurons.

### Endogenous protein labeling in specific tissues

We have developed a reliable method to label endogenous proteins in specific tissues using split GFP. Our studies highlight the importance of maintaining ribosomal protein expression at levels close to endogenous. *C. elegans* ribosome biogenesis seems to be tightly regulated so that the total amount of ribosomal proteins is balanced as in *E. coli* (*Kaczanowska and Rydén-Aulin, 2007*) because copy numbers of endogenous *rps-18* or *rpl-29* affected the expression level of single-copy-inserted GFP-tagged ribosomal proteins. This regulation likely exists because imbalanced expression of ribosomal proteins results in cellular defects, as we showed using high-copy transgenes. Overexpression of tagged proteins can also cause signal artifacts, which are a common concern for

**Table 1.** Mutants with abnormal ribosome distributions

| Gene | Protein | Allele | NT change | AA change | Inheritance | Primers | Note |
|------|---------|--------|-----------|-----------|-------------|---------|------|
| *mec-7* | β-tubulin | *ju1286* | G722A | G144R | Recessive | YJ11867/YJ12057 | Exchangeable GTP/GDP-binding site |
| *mec-7* | β-tubulin | *ju1287* | C705T | S138L | Recessive | YJ11867/YJ12057 | Exchangeable GTP/GDP-binding site |
| *mec-7* | β-tubulin | *ju1294* | G717A | G142D | Recessive | YJ11867/YJ12057 | Exchangeable GTP/GDP-binding site |
| *mec-7* | β-tubulin | *ju1298* | G603A | G104E | Recessive | YJ11867/YJ12057 | Exchangeable GTP/GDP-binding site |
| *mec-7* | β-tubulin | *ju1297* | G121A | R2H | Dominant | YJ12136/YJ12137 | Non-exchangeable GTP-binding site |
| *mec-12* | α-tubulin | *ju1295* | G7197A | G354E | Dominant | YJ12134/YJ12135 | Non-exchangeable GTP-binding site |
| *mec-15* | E3 ligase | *ju1288* | G327A | W91X | Recessive | YJ11668/YJ11669 | |
| *mec-15* | E3 ligase | *ju1289* | G217A G1661A | (Splice acceptor) R342Q | Recessive | YJ11668/YJ11669 YJ12138/YJ12139 | |

DOI: https://doi.org/10.7554/eLife.26376.015

visualizing proteins in live cells. For example, to avoid the artifact by overexpressing PSD-95, a post-synaptic protein (*El-Husseini et al., 2000*), recombinant antibody-like proteins called FingRs have been developed (*Gross et al., 2013*). Tissue-specific expression of GFP-tagged FingRs can visualize an endogenous protein without tagging. However, this method requires the generation of optimal FingRs with mRNA display; moreover, excess GFP-FingRs can produce background signals. The advantage of split GFP is that neither GFP1-10 nor GFP11 is fluorescent. In contrast to antibody-based methods, however, the split GFP method requires genetic alteration of the endogenous locus. With recent advances in genome engineering with the Clustered Regularly Interspaced Short Palindromic Repeats (CRISPR) method, it is feasible to tag the endogenous locus with a fluorescent protein in various organisms (*Hsu et al., 2014*). To avoid interfering with the function of endogenous proteins, it is desirable to keep the tag as small as possible. GFP11 satisfies this requirement because it has only 16 amino-acids. Endogenous protein labeling brings another challenge in that widespread endogenous expression in many tissues might prevent visualization in a tissue of interest, as is the case for the ribosomal proteins. GFP11 can also provide tissue specificity.

To maximize RIBOS signals, we found it necessary to introduce the *rps-18(0)* background and to saturate GFP11::RPS-18 with excess GFP1-10. The saturation could be achieved by using multi-copy GFP1-10 transgenes because we found that decreasing the copy number of integrated GFP1-10 transgenes did not change the intensity of RIBOS signals (data not shown). Given that *rps-18* is an abundant ribosomal protein, we expect that we should be able to saturate most GFP11-tagged endogenous proteins using multi-copy expression of GFP1-10. To further increase the signals, a tandem GFP11 tag might be used. This is a similar idea to the SunTag approach, in which fluorescently tagged antibodies recognize tandem peptide tags in a targeted protein (*Tanenbaum et al., 2014*). A tandem split GFP approach may have an advantage over SunTag because an excess amount of split GFP does not produce background, unlike fluorescently tagged antibodies. Recently, split GFP has been used as a versatile tool for endogenous protein visualization in cultured cells (*Kamiyama et al., 2016*). In multicellular organisms, it provides the additional advantage of tissue- or cell-specificity as demonstrated in this study. Thus, the split-GFP-based approach can be broadly applied to visualize any endogenous protein in specific cells.

## Dynamic and cell-specific ribosome localization

In non-neuronal tissues, ribosomes existed throughout the cytosol even in a fine structure like muscle arms. In contrast, ribosomes were mostly restricted in the soma in neurons. Moreover, neuronal ribosomes showed distinct localization, such as at the branch point of PLM mechanosensory neurons and at presynaptic terminals of motor neurons at the L1 stage animals. Moreover, we found that ribosome localization changed during development. These distribution patterns were different from other protein translation-related components, such as initiation factors and RNA granules. Together, our results suggest that neuronal ribosome localization is actively regulated.

Using electron microscopy, we have further demonstrated the presence of ribosomes at the presynaptic terminals of motor neurons in adult *C. elegans*. In contrast to postsynapses and axons, the evidence of the existence of ribosomes at presynaptic terminals has been limited. However, some studies have shown the functional importance of local translation at presynaptic terminals (*Beaumont et al., 2001*; *Martin et al., 1997*). In squid synaptosomal fractions and optic nerve endings, ribosomes have been observed by electron energy loss spectroscopic microscopy and conventional EM (*Crispino et al., 1997*; *Martin et al., 1998*). In *C. elegans* ribosome-like black dots were occasionally observed in the presynaptic terminals in the ventral nerve cord (*Rolls et al., 2002*). Our study provided more definitive morphological evidence for the existence of ribosomes at mature presynaptic terminals by using electron tomography, in which three-dimensional morphologies are more clearly visualized than conventional EM studies. RIBOS markers only visualized ribosomes in the presynaptic terminals at the L1 stage but not at the adult stage, probably because the low density (a few ribosomes per synapse) was below the detection limit for a fluorescent protein.

Since a low abundance of ribosomes at presynaptic terminals could not be detected by RIBOS, the punctate signals observed along the mechanosensory axon were likely to be clusters of ribosomes. This cluster resembles the plaque-like ribosomes in the vertebrate axons (*Koenig, 2009*), suggesting that clustering of ribosomes may be a common feature throughout evolution. The ribosome puncta did not seem to be associated with transported vesicles because they were static over periods of a few minutes. However, ribosome distribution changed during development and after

axon injury. This suggests that ribosome localization may rely on slow axonal transport, known to control the distribution of many cytoskeletal components and cytosolic proteins (*Brown, 2003*).

In *C. elegans* mechanosensory neurons, mRNA of a transcription factor, CCAAT/enhancer-binding proteins (*cebp-1*), is localized to the axon and locally translated in an axon-injury-dependent manner (*Yan et al., 2009*). Consistent with this, we observed RIBOS signals and monosomes in the electron micrographs after axon injury. Although monosomes have been thought to be translationally inactive, recent ribosome profiling data suggest that they can be active (*Heyer and Moore, 2016*). In rat dentate gyrus, the number of the spines associated with polysomes increased several days after lesion (*Steward, 1983*). Increased number of axonal ribosomes after injury is also shown in rat motor axons of ventral spinal roots (*Zheng et al., 2001*) and sciatic nerve axons in the Wallerian degeneration slow mouse (*Court et al., 2008*). These studies highlight that the dynamic nature of ribosome distribution after nerve injury may be conserved throughout evolution.

In this study, we focused on RPS-18 as a representative protein for the ribosome function because we could rely on the rescue of the *rps-18(0)* mutant phenotypes as a proxy to assess the *in vivo* activity of the functional protein. Recent work in mice has shown tissue-type heterogeneity of ribosome composition (*Simsek et al., 2017*). Rpl38 knock-out mice show tissue-specific patterning defects, due to selective reduction on translation of mRNAs for some Hox genes (*Kondrashov et al., 2011*; *Xue et al., 2015*). It is possible that in *C. elegans* not all ribosomes contain RPS-18, and therefore, that our approach may tag only a subset of ribosomes. Future studies will improve and extend the RIBOS approach to address this issue by systematically tagging different ribosomal proteins with GFP11 using CRISPR-based knock-in to endogenous locus of ribosome genes.

## Ribosomes distribution is regulated by *unc-16* and microtubules

Through candidate and forward genetic screens, we have identified UNC-16 and microtubules as regulators of ribosome distribution in *C. elegans* mechanosensory neurons.

UNC-16 is a member of the conserved JIP3 proteins (*Byrd et al., 2001*). These proteins act as an adaptor between cargos, for example mitogen-activated protein kinase (MAPK) components, and kinesin (*Koushika, 2008*; *Whitmarsh, 2006*). In *C. elegans*, *unc-16* was originally isolated by its uncoordinated movement and later shown to display mislocalized synaptic vesicles (*Byrd et al., 2001*). In cholinergic motor neurons, UNC-16 was found to restrict membranous organelles, such as Golgi and endosomes to cell bodies (*Edwards et al., 2013*). Our data show that UNC-16 regulates distribution of a large molecular complex, ribosomes. In contrast to the increased axonal endosomes and Golgi in *unc-16* mutants, ribosomes were increased in the presynaptic terminals and decreased along the proximal axon. Since UNC-16 localizes to the proximal axon in mechanosensory neurons (*Edwards et al., 2013*), these results might imply that UNC-16 restricts the ribosomes into the soma at the proximal axon.

Although both ribosomes and microtubules are well characterized cellular components, the connection between them has not been well appreciated. Both EM studies and biochemical evidences show that ribosomes can interact with microtubules (*Suprenant, 1993*; *Suprenant et al., 1989*). Microtubule disruption by nocodazole treatment makes ribosomes move into the neurites in chicken sensory neuronal cultures (*Baas et al., 1987*). We observed ribosomes at the cutting site after axon injury. This might be because an altered state of microtubule after axon injury (*Chisholm, 2013*) allowed ribosomes to move into the axon. In *C. elegans* mechanosensory neurons, the number of growing microtubules increases three hours after axon injury, followed by consistent microtubule growth (*Ghosh-Roy et al., 2012*). We also observed fewer numbers of microtubules at the injury site by EM. Mitochondria observed at the cutting site may locally supply energy for translation and microtubule dynamics. These data are consistent with a hypothesis that microtubules restrict the ribosomes into the neuronal soma in unstressed conditions in mature neurons. We observed that ribosome distribution in the soma was also disrupted in the microtubule mutants. In neuronal soma, ribosomes are dynamically exchanged between two states: free in the cytosol or bound to the ER (*Jan et al., 2014*). This compartmentalized protein synthesis ensures the efficient targeting of transmembrane and secreted proteins to the ER. The balance between ER-bound and free ribosomes is important because in nascent polypeptide-associated complex (NAC) mutants in *C. elegans*, ribosomes are mistargeted to the ER, resulting in disrupted protein homeostasis (*Gamerdinger et al., 2015*). Mislocalization of ribosomes in microtubule mutants may suggest that microtubules are components to prevent mistargeting ribosomes in the soma.

## Materials and methods

### Transgene design and rescue

To label ribosomal proteins, we chose several candidates whose N- or C- terminus is exposed to the surface of ribosomes based on the eukaryotic ribosome structure (*Ben-Shem et al., 2011*). We first generated multi-copy transgenic lines for *rpl-1*, *rpl-4*, *rpl-29*, *rpl-30* and *rps-18*. Homologs of *rpl-1* and *rpl-4* were used for labeling ribosomes in mice (*Court et al., 2008*; *Heiman et al., 2008*). However, the overexpression of these proteins in mechanosensory neurons caused abnormal axon and soma morphologies. We suspect that such defects could be due to overexpression. We then focused on *rpl-30* and *rps-18*, null mutations in which caused larval arrest. Extrachromosomal arrays over-expressing N-terminally tagged *rps-18* partially rescued *rps-18(0)* larval arrest, whereas C-terminally tagged *rps-18* or *rpl-30* did not. In addition, we used *rpl-29* as a component of the ribosome large subunit; *rpl-29* null mutants (*tm3555*) did not have any behavioral or morphological phenotypes.

### Plasmids and strains

*C. elegans* strains were maintained as previously described on Nematode Growth Medium (NGM) plates at 22.5°C unless otherwise noted (*Brenner, 1974*). Strains, plasmids, and primers are summarized in *Supplementary files 1*, *2,* and *3*, respectively. Plasmids were constructed using standard techniques including restriction enzyme digestion and ligation, QuikChange site-directed mutagenesis (Agilent Technologies, Santa Clara, CA), Gateway cloning (Invitrogen, Carlsbad, CA), and Gibson assembly (NEB). An *rps-18* genomic fragment including 567 bp upstream and 162 bp downstream of the coding sequence was cloned into pCR8 vector (Invitrogen). GFP was inserted using the BglII and PstI sites to generate the GFP::*rps-18* vector, which contains a three amino-acid (AAG) linker between GFP and RPS-18. GFP::*rps-18* in pCR8 vector was recombined with a pCFJ150-based MosSCI backbone vector to generate pCZGY3178 using Gateway LR clonase (Invitrogen). pCZGY3178 plasmid was injected at 50 ng/µl with other plasmids into EG4322 *ttTi5605; unc-119 (ed9)* to generate a single-copy-inserted transgene *juSi83[GFP::rps-18+Cb-unc-119(+)]* II using a standard MosSCI insertion method (*Frøkjaer-Jensen et al., 2008*). GFP11 was cloned from MVC3 (pSM_nlg-1_GFP11) and inserted into the genomic *rps-18* in pCR8 vector using the BglII and PstI sites to generate pCZGY3163, which contains 15 amino-acid linker (ELGSGSGSGSGSSAG) between GFP11 and RPS-18. GFP11::*rps-18* from pCZGY3163 was cloned into MosSCI vector with a multi cloning site (pCFJ151) to generate pCZ946. pCZ946 plasmid was used to generate a single-copy-inserted transgene *juSi94[GFP11::rps-18 + Cb-unc-119]* II as described above. RIBOS strains were made by injecting the corresponding plasmids to CZ17515 *juSi94; rps-18(ok3353)* using P*ttx-3*-RFP as a co-injection marker labeling AIY neurons (see *Supplementary file 1*). Integrated transgenic lines were made using the UV/TMP method and outcrossed several times before analyses.

### Fluorescence microscopy and FRAP

For imaging or scoring phenotypes or expression pattern with microscopes, worms were immobilized with 25%(v/v) of 50 nm polystyrene beads (Polysciences, Inc.) in M9. Ribosome defects in mechanosensory neurons were visualized by P*mec-4*-RIBOS in mutant backgrounds and scored under a Zeiss Axioplan 2 microscope equipped with Chroma HQ filters and a 63x (NA=1.4) objective lens. For drug treatment, colchicine plates were prepared by spreading 1 mM colchicine (Sigma) solution on NGM plates and spotting concentrated OP50 bacteria. For developmental time course experiment, worms were synchronized by letting parental worms lay eggs for one hour and keeping the progeny at 20°C. Representative images of fluorescent reporters except for *juSi83* and *juSi123* were collected using a Zeiss LSM710 confocal microscope equipped with a 63x (NA=1.4) or 100× (NA=1.46) objective lens. *juSi83[GFP::rps-18]* and *juSi123[GFP::rpl-29]* strains were imaged using a Zeiss Axioplan 2 microscope equipped with Chroma HQ filters and a 10x (NA=0.3) objective lens. The zoom function of Zeiss ZEN software was used for the mechanosensory neuron cell body and FRAP. Images shown are single-plane images or maximum-intensity-projections obtained from a few z-sections (0.5 µm/section) using Zeiss ZEN software. Note that signals in the cell body were saturated to take images of subcellular components, such as muscle arms and axons. Without GFP11::*rps-18*, faint fluorescence due to GFP1-10 was detected in some nuclei and was negligible compared

to the reconstituted GFP signals in RIBOS. Intensities were analyzed after subtracting background signals using FIJI software(*Schindelin et al., 2012*).

FRAP analysis was performed using an LSM710 with the 63x objective lens (NA=1.4) and young adult worms. For the photobleach, 10 and 5 scanning iterations were performed for a 1 μm region of interest in the basal region of epidermis and ALM cell body, respectively. Time series of a single plane were collected at 5 s and 0.4 s intervals for ribosomes and initiation factors, respectively. Percent of recovery after photobleaching was calculated after background subtraction and correction for fluorescence loss during image acquisition. The data points were plotted and fitted with one-phase decay function to calculate half recovery time ($t_{1/2}$) and mobile fractions using GraphPad Prism v5.0 software (GraphPad Software, La Jolla, CA).

## Axotomy

Axotomy experiments were performed by cutting axons 40–50 μm away from the soma using a near infrared Ti-Sapphire laser as previously described (*Wu et al., 2007*). For cycloheximide treatment, worms were kept on NGM plates without drugs before axotomy, and PLM axons visualized by *muIs32[Pmec-7-GFP]* or *zdIs5[Pmec-4-GFP]* were severed at the L4 stage. Worms were recovered on the NGM plates containing 1 mM cycloheximide or DMSO, and imaged 24 hr after axotomy using LSM710 as described above. Regrowth was calculated by subtracting the axonal length immediately after axotomy from the length 24 hr after axotomy using FIJI software (*Schindelin et al., 2012*). For RIBOS analysis after axotomy, ALM axons visualized by P*mec-4*-tagRFP in *juIs438* were severed, imaged for RIBOS signals using the LSM710, and analyzed using FIJI software.

## Electron microscopy for serial reconstructions

L1 or young adult N2 worms were used for analyzing motor neurons. CZ10175 *zdIs5[Pmec-4-GFP]* worms were axotomized at the L4 stage as described above and used six hours after axotomy. Worms were frozen under high pressure at −176°C with a high-pressure freezer HPM 010 (BAL-TEC) and then freeze substituted in 2% osmium tetroxide and 0.1% uranyl acetate in acetone for 4 d at −90°C and for 16 hr at −20°C with a freeze-substitution apparatus EM AFS2 (Leica). After infiltration and embedding in Durcupan ACM resin blocks were polymerized at 60°C for 48 hr. Fifty nanometer-thick serial sections were cut with a diamond knife from anterior part of worm after posterior pharyngeal bulb with an ultramicrotome ULTRACUT UCT (Leica), collected onto pioloform coated slot grids and stained for 5 min in 70% methanol with 2.5% uranyl acetate, followed by washing for 3 min in Reynold's lead citrate. Serial images from dorsal and ventral nerve cords were collected with a digital camera with 2688×2672 pixel resolution (Gatan) and DigitalMicrograph acquisition software (Gatan) on a transmission electron microscope JEOL-1200 EX (JEOL) at magnification of 10,000 × at an accelerating voltage of 80 kV. All digital images were imported into the series with Reconstruct 3D reconstruction software and realigned for accurate 3D measurements and visualizations. Ribosomes were defined as electron-dense objects with an approximately 20 nm diameter, which are not continuous on serial sections. Information about membranes, microtubules, mitochondria, and ribosomes, was created by manually tracing the profiles of the same objects on the serial image sections with Graphire3 Pen Tablet input hardware (Wacom). After tracing the profiles on the sections, the 3D scenes were rendered and saved as a 360-degree bitmap images.

## Electron tomography

Resin blocks containing N2 young adult worms were prepared as described above. The 300–350 nm-thick sections from anterior part of worm were cut with a diamond knife and an ultramicrotome ULTRACUT UCT (Leica), collected onto copper folding mesh grids without a supporting film, and stained on both sides for 10 min in 70% methanol with 2.5% uranyl acetate, followed by washing for 5 min in Reynold's lead citrate. Colloidal gold particles (15 nm in diameter) were deposited on both sides of the sections for 5 min to serve as fiducial cues. For stability in the beam, the sections were coated with carbon on both sides. Series of images from synapses in the ventral nerve cord were collected with a digital camera with 3488×3488 pixel-resolution (GATAN) and Serial EM acquisition software on an intermediate-voltage electron microscope JEOL-4000 EX (JEOL) at magnification of 25,000×at an accelerating voltage of 400 kV. To minimize shrinkage of the sections during the tilt series and reduce the effects of within-tilt-series variation on the subsequent back projection, the

sections were pre-irradiated for 15 s at each tilt angle before image acquisition. Series of 61 images were acquired by rotating the sample from −60 to +60 degrees around an axis perpendicular to the optical axis of the microscope in 2-degree increment with a computer-controlled goniometer. The grid was then rotated 90 degrees in plane and another tilt series was collected. Pixel size of the tilt series acquisition images was 0.607 nm per pixel. The tilt-series images were aligned using IMOD software. Then, IMOD or TxBR software was used to create the final alignments and back projections resulting in three-dimensional volumes (tomograms). The tomograms were opened in IMOD software and saved as snapshot images. Objects were traced by the same method as for serial reconstruction.

### Genetic screen and mapping

The forward genetic screens to look for ribosome distribution defects were carried out using P*mec-4*-RIBOS strain with *glo-4* mutant background (CZ19298), which was introduced for better visualization of the fluorescent signals for the screen by reducing gut autofluorescence (*Hermann et al., 2005*). After treating worms with 50 mM EMS using a standard protocol (*Brenner, 1974*), one or four $F_1$ animals were transferred to individual plates. $F_2$ animals were mounted on slides using M9 and subjected to visual inspection under the Zeiss Axioplan 2 as described above. Eight mutant worms were recovered from 1400 haploid genomes. After determining inheritance and complementation, we mapped mutants by sequencing known mechanosensory (*mec*) genes.

### Statistical analysis

We used Student's t-test for comparisons of two samples, and one-way ANOVA with Bonferroni's multiple comparison test for the others in GraphPad Prism 5.0 (GraphPad Software, La Jolla, CA)

## Acknowledgements

We thank the Caenorhabditis Genetics Center (CGC), National BioResource Project and Dr. S Mitani for strains, and WormBase for resources. We are grateful to AD Chisholm and our lab members for valuable input throughout the work.

## Additional information

### Funding

| Funder | Grant reference number | Author |
|---|---|---|
| Howard Hughes Medical Institute | | Yishi Jin |
| National Institutes of Health | R01 035546 | Yishi Jin |

The funders had no role in study design, data collection and interpretation, or the decision to submit the work for publication.

### Author contributions

Kentaro Noma, Conceptualization, Data curation, Formal analysis, Investigation, Methodology, Writing—original draft; Alexandr Goncharov, Data curation, Formal analysis, Methodology; Mark H Ellisman, Resources, Funding acquisition; Yishi Jin, Conceptualization, Resources, Supervision, Funding acquisition, Writing—original draft, Project administration

### Author ORCIDs

Kentaro Noma http://orcid.org/0000-0002-6487-8037
Yishi Jin http://orcid.org/0000-0002-9371-9860

### Decision letter and Author response

Decision letter https://doi.org/10.7554/eLife.26376.020
Author response https://doi.org/10.7554/eLife.26376.021

## Additional files

### Supplementary files

• Supplementary file 1. Strain list.
DOI: https://doi.org/10.7554/eLife.26376.016

• Supplementary file 2. Plasmid list.
DOI: https://doi.org/10.7554/eLife.26376.017

• Supplementary file 3. Primer list.
DOI: https://doi.org/10.7554/eLife.26376.018

• Transparent reporting form
DOI: https://doi.org/10.7554/eLife.26376.019

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
