## [Decision Letter]

Thank you for submitting your article "Microtubule-dependent ribosome localization in *C. elegans* neurons" for consideration by *eLife*. Your article has been reviewed by three peer reviewers, and the evaluation has been overseen by Anna Akhmanova as the Senior Editor and Reviewing Editor. The reviewers have opted to remain anonymous.

The reviewers have discussed the reviews with one another and the Reviewing Editor has drafted this decision to help you prepare a revised submission.

Summary:

In their manuscript, Yishi Jin and colleagues introduce an elegant method to observe ribosome localization in vivo: RIBOS. In RIBOS, one GFP β-sheet is fused to a ribosomal protein subunit and complemented with the remainder of the split GFP, which is expressed in a tissue-specific manner. The observed axonal localizations are validated using electron microscopic tomography. The authors demonstrate the usefulness of their approach by studying axon injury, which recruits ribosomes as expected, and by characterizing transport factors necessary for ribosomal localization in a small candidate screen, as well as an unbiased EMS screen. While the latter can certainly be enlarged – which would go beyond the scope of this paper – it clearly shows the power of the approach. Importantly, evidence of ribosome localization in axons of mature brain has been limited and therefore this manuscript indirectly strengthens the importance of axonal protein synthesis in matured axons. In addition, the authors provide evidence that microtubules might be important for the generation of a barrier in the proximal axon.

This is an original method that meets a demand for techniques to study ribosome localization in vivo. The experiments are carefully conducted and analyzed, and the Materials and methods section is detailed enough to allow other researchers to adopt RIBOS for their work.

Essential revisions:

1) The authors use only indirect methods to demonstrate that GFP11::RPS-18 is incorporated into ribosomes. A biochemical approach, such as polysome profiling, should be performed to answer this question more directly, and to determine how much (if any) GFP11::RPS-18 is or is not part of ribosomes, leading to false positive RIBOS signals.

Furthermore, the authors need to at least discuss one potential pitfall of the method: in the target cells, the investigated ribosomal protein is irreversibly reconstituted to contain a full GFP tag. The sterility of the GFP-tagged strain suggests that the fusion protein does not function for some special purposes. Therefore, it is possible that some (local?) translation is inhibited also by the reconstituted GFP fusion, or that ribosomal protein molecules that have been recruited to that particular function cannot be reconstituted with GFP, i.e., this special translation cannot be observed.

2) Ribosome composition is regulated developmentally and differs between tissues. Any method that relies on ribosomal protein tagging will necessarily only label a subset of ribosomes. An analysis of how many ribosomes in the studied *C. elegans* neurons contain RPS-18 is needed to determine how representative for total -as opposed to RPS-18 containing- ribosome localization RIBOS is. The same issue should at least be discussed in relation to the reported developmental and injury induced changes in the axonal RIBOS signal.

3) Although differences in recovery times of the FRAP experiments (Figure 1/G) seem obvious, statistical analysis should be provided to assess those differences. Additionally, recovery curves in Figure 1 should be plotted together to better appreciate the difference between rpS18 and eIF4E (IFE-2 and IFE-4) mobility. Finally, it needs to be discussed that the mobile fraction/phase is rather different between rpS18 and eIF4E.

4) In Figure 4, panels A and C show only one of the developmental stages analyzed. The authors should provide representative images of both larva and adult stages for comparison.

5) In Figure 5 there is an increase in RIBOS signal distal to the injury site at 12 h (pointed out with an arrow in 5). Is this the regrowing axon – the text states that the axons regrow only at 24h – or is it the severed distal part of the axon?

6) Statistical analysis for Figure 7 should be added.

7) The increase in co-localisation in Figure 7 is not so evident. If the authors think this point is important (I think it can be deleted without any important loss for the paper), the% co-localization should be quantified.

8) The decrease in fluorescence intensity of the free RFP cannot be taken as an indication of reduced translation (subsection “A forward genetic screen identified roles for tubulins in ribosome distribution”, last paragraph; Figure 7). There are too many other factors influencing this read-out. Please revise the text accordingly.

---

## [Author Response]

*Essential revisions:*

*1) The authors use only indirect methods to demonstrate that GFP11::RPS-18 is incorporated into ribosomes. A biochemical approach, such as polysome profiling, should be performed to answer this question more directly, and to determine how much (if any) GFP11::RPS-18 is or is not part of ribosomes, leading to false positive RIBOS signals.*

We agree with the reviewers that our evidence for GFP11::RPS-18 incorporation into ribosomes is indirect. We had planned to perform polysome profiling and detect RPS-18 distribution by Western blotting. However, antibodies recognizing *C. elegans* RPS-18 are not available. We found two commercially advertised anti-human RPS-18 antibodies; unfortunately, the delivery of these antibodies was back ordered, and we received them only a week ago. At this moment, we do not yet know if these antibodies could be proven to have specificity for *C. elegans* RPS-18. Additionally, we are not aware of the feasibility of tissue-specific isolation of polysomes from *C. elegans.* Therefore, we cannot estimate the time required and outcome of the polysome profiling experiment.

*Furthermore, the authors need to at least discuss one potential pitfall of the method: in the target cells, the investigated ribosomal protein is irreversibly reconstituted to contain a full GFP tag. The sterility of the GFP-tagged strain suggests that the fusion protein does not function for some special purposes. Therefore, it is possible that some (local?) translation is inhibited also by the reconstituted GFP fusion, or that ribosomal protein molecules that have been recruited to that particular function cannot be reconstituted with GFP, i.e., this special translation cannot be observed.*

We agree with the comments. It is possible that reconstituted GFP11::RPS-18 and GFP1-10 may not be fully functional. We revised text as suggested:

“Since the binding between GFP11 and GFP1-10 is irreversible, it is possible that the reconstituted GFP::RPS-18 is not fully functional in contexts where ribosome activity is locally required or sensitive to dosage.”

*2) Ribosome composition is regulated developmentally and differs between tissues. Any method that relies on ribosomal protein tagging will necessarily only label a subset of ribosomes. An analysis of how many ribosomes in the studied C. elegans neurons contain RPS-18 is needed to determine how representative for total -as opposed to RPS-18 containing- ribosome localization RIBOS is. The same issue should at least be discussed in relation to the reported developmental and injury induced changes in the axonal RIBOS signal.*

We indeed thought about tissue- and developmental-dependency of ribosome composition when developing our method. We had surveyed the expression pattern for several ribosomal protein genes, such as *rpl-1, rpl-12, rpl-14*, and *rpl-15*; we found that like *rps-18* and *rpl-29,* these ribosome genes were ubiquitously expressed in all somatic tissues. These observations are consistent with published information on other ribosomal genes from RNAi knockdown studies. The apparent ubiquitous expression of GFP::RPS-18 and the lethal and sterile phenotypes of *rps-18(0)* mutants are indications that RPS-18 is likely to be a component of most, if not all, ribosomes. We agree that some types of ribosomes could contain specific proteins, or that some ribosomal proteins may only be essential in certain cell types. We believe that our present study provides a starting point for such issues to be addressed in vivo. In the revised manuscript, we have included following discussion:

“In this study, we focused on RPS-18 as a representative protein for the ribosome function because we could rely on the rescue of the rps-18(0) mutant phenotypes as a proxy to assess the in vivo activity of the functional protein. […] Future studies will improve and extend the RIBOS approach to address this issue by systematically tagging different ribosomal proteins with GFP11 using CRISPR-based knock-in to endogenous locus of ribosome genes.”.

*3) Although differences in recovery times of the FRAP experiments (Figure 1/G) seem obvious, statistical analysis should be provided to assess those differences. Additionally, recovery curves in Figure 1 should be plotted together to better appreciate the difference between rpS18 and eIF4E (IFE-2 and IFE-4) mobility. Finally, it needs to be discussed that the mobile fraction/phase is rather different between rpS18 and eIF4E.*

In the revised Figure 1, we combined previous Figure 1 as Figure 1, and provided the statistical analysis for the comparison of t_1/2_ (Figure 1) and mobile fraction (Figure 1). The figure legend and main text are revised accordingly.

“If GFP11::RPS-18 is incorporated into a large organelle or complex, such as the ribosome, the half time of fluorescence recovery (t_1/2_) should be larger and the mobile fraction should be smaller than those of a free cytosolic protein (Lippincott-Schwartz et al., 2001)”

“Photobleached GFP11::RPS-18 showed a longer recovery time and smaller mobile fraction than IFE-2::GFP and IFE-4::GFP (Figure 1). Furthermore, the recovery curve, t_1/2_, and mobile fraction of GFP11::RPS-18 were comparable to that of the ribosomal large subunit, RPL-29::GFP (Figure 1), suggesting that fluorescence recovery was not affected by splitting GFP, and that RPS-18 and RPL-29 were incorporated into a complex with a similar mobility, likely the ribosome.”.

Legends for Figure 1, and 1H:

“(F) Fluorescence recovery after photobleaching was plotted for juEx5809[Pcol-19-IFE-2::GFP], *juEx5811[Pcol-19-IFE-4::GFP],* Pcol-19-RIBOS, and *juSi123[RPL-29::GFP]; rpl-29(0)*. The line represents the one-phase fit to an exponential function for each plot. The inset shows the magnified graph for IFE-2::GFP and IFE-4::GFP. Error bars indicate S.E.M. (G and H) t_1/2_ and mobile fraction calculated from (F). Error bar indicates S.E.M., Statistics: One-way ANOVA, ns: p>0.05, p**<0.01, p****<0.0001.”

*4) In Figure 4, panels A and C show only one of the developmental stages analyzed. The authors should provide representative images of both larva and adult stages for comparison.*

We have included an RIBOS image at the adult stage in panel B (previously panel C) for comparison. We did not include an L1 image to Panel A, because this panel is for comparing RIBOS and the negative control.

*5) In Figure 5 there is an increase in RIBOS signal distal to the injury site at 12 h (pointed out with an arrow in 5). Is this the regrowing axon – the text states that the axons regrow only at 24h – or is it the severed distal part of the axon?*

The arrow in Figure 5 is showing the tip of regrowing axon 12 hours after axotomy, which could be distinguished from the distal axon based on the z-plane. The regrowth starts about 6 h after injury. We clarified this point in the revised text.

“Around 6 hours after axotomy, microtubules are reorganized, leading to the formation of a growth cone, and severed axons start regrowing around the same time, resulting in average regrowth about ~100 µm after 24 hours (Chen et al., 2011; Ghosh-Roy et al., 2012).”

“We also observed weak RIBOS signals in the regrowing tip (Figure 5).”

*6) Statistical analysis for Figure 7 should be added.*

We added statistical analyses using Fisher’s exact test in the revised Figure 7. We deleted Figure 7 as discussed below in 7 and 8.

*7) The increase in co-localisation in Figure 7 is not so evident. If the authors think this point is important (I think it can be deleted without any important loss for the paper), the% co-localization should be quantified.*

*8) The decrease in fluorescence intensity of the free RFP cannot be taken as an indication of reduced translation (subsection “A forward genetic screen identified roles for tubulins in ribosome distribution”, last paragraph; Figure 7). There are too many other factors influencing this read-out. Please revise the text accordingly.*

We deleted the paragraph discussing preferential ribosome localization to the ER in the previous manuscript including Figure 7 and Figure 7—figure supplement 3). Our work does not attempt to address whether there is preferential localization of ribosomes in the ER, but focuses on a new method to visualize ribosomes in live animals. We consider possible ER localization in the revised Discussion.

“We observed that ribosome distribution in the soma was also disrupted in the microtubule mutants. […] Mislocalization of ribosomes in microtubule mutants may suggest that microtubules are components to prevent mistargeting ribosomes in the soma.”